# Impact of Neoadjuvant Treatment on Body Composition in Patients with Locally Advanced Gastric Cancer

**DOI:** 10.3390/cancers16132408

**Published:** 2024-06-29

**Authors:** Luz Divina Juez, Pablo Priego, Marta Cuadrado, Luis A. Blázquez, Silvia Sánchez-Picot, Pablo Gil, Federico Longo, Julio Galindo, José María Fernández-Cebrián, José I. Botella-Carretero

**Affiliations:** 1Department of General and Digestive Surgery, Hospital Universitario Ramón y Cajal, 28034 Madrid, Spain; luzdivina.juez@salud.madrid.org (L.D.J.); julio.galindo@salud.madrid.org (J.G.);; 2Instituto Ramón y Cajal de Investigación Sanitaria, IRyCIS, Hospital Universitario Ramón y Cajal, 28034 Madrid, Spain; 3Faculty of Medicine, University of Alcalá (UAH), Alcalá de Henares, 28801 Madrid, Spain; 4Department of General and Digestive Surgery, Hospital Universitario La Paz, 28046 Madrid, Spain; 5Department of Clinical Oncology, Hospital Universitario Ramón y Cajal, 28034 Madrid, Spain; 6Department of Endocrinology and Nutrition, Hospital Universitario Ramón y Cajal, 28034 Madrid, Spain

**Keywords:** sarcopenia, neoadjuvant, gastric, cancer, gastrectomy

## Abstract

**Simple Summary:**

Neoadjuvant chemotherapy (NT) followed by radical surgery is the standard treatment for locally advanced gastric cancer (GC). The impact of NT on body composition was analyzed in locally advanced GC patients undergoing gastrectomy. During chemotherapy, GC patients experienced loss of muscle mass and fat, leading to an increased incidence of pre-surgical sarcopenia.

**Abstract:**

Neoadjuvant chemotherapy (NT) followed by radical surgery is the standard treatment for locally advanced gastric cancer (GC). The incidence of sarcopenia in upper gastrointestinal tract malignancies is very high, and it may be increased after NT. This study aimed to evaluate the impact of NT on body composition. A retrospective study of patients with locally advanced GC undergoing gastrectomy who had received NT in a tertiary hospital between 2012 and 2019 was conducted. CT measured the skeletal muscle index, total psoas area, and visceral and subcutaneous adipose tissue before and after NT. Of the 180 gastrectomies for GC, 61 patients received NT. During NT, changes in body composition were observed with a decrease in the skeletal muscle mass index (SMMI −2.5%; *p* < 0.001), and these changes were significantly greater in men (SMMI −10.55%). Before surgery, patients who received NT presented 15% more sarcopenia than those without NT (*p* = 0.048). In conclusion, patients with locally advanced gastric cancer who receive NT have significant changes in body composition during chemotherapy. These changes, which are at the expense of a loss of muscle mass, lead to an increased incidence of pre-surgical sarcopenia.

## 1. Introduction

Gastric cancer (GC) is the fifth most common cancer around the world, with an estimated 783,000 deaths in 2018 the world [1]. Unfortunately, despite medical advances, GC in Western countries is often diagnosed in advanced stages. Historically, radical surgery has been the only curative treatment for gastric cancer. In 2006, Cunningham et al. published the results of the MAGIC study. Their results showed an improvement in overall survival (23% vs. 36%) and 5-year disease-free survival for gastric cancer patients receiving a perioperative chemotherapy regimen [2]. Currently, even though the chemotherapy treatment regimen has recently been modified due to the findings of Al-Batran [3], neoadjuvant chemotherapy (NT) followed by radical surgery is the optimal approach for locally advanced GC [4,5].

Many factors contribute to the prognosis of patients with gastric cancer, and one of the most important is their nutritional status. In GC, the prevalence of malnutrition is even higher than in other tumors, estimated at around 60% [6]. According to the European Society for Clinical Nutrition and Metabolism (ESPEN), disease-related malnutrition results from the activation of systemic inflammation by an underlying disease, which in this case is an oncological process [7]. Furthermore, the modern term sarcopenia, defined as a syndrome characterized by a progressive and widespread loss of skeletal muscle mass and strength, is associated with a physical disability, poorer quality of life, and increased mortality [8]. The incidence of sarcopenia in patients with upper gastrointestinal tract neoplasms is very high [9]. Some authors strongly correlate the presence of sarcopenia with more postoperative complications and worse overall survival (OS) in patients with GC [10,11,12].

As part of the oncological staging study, all patients diagnosed with GC undergo a computed tomography (CT) scan. Radiological reports usually only reflect data related to the oncological aspects of the tumor [13]. We have recently shown the impact of body composition evaluated by CT scan on oncologic outcomes and postoperative complications following gastrectomy [14]. In this study, patients with sarcopenia and sarcopenic obesity exhibited poorer overall survival and increased severe postoperative morbidity. However, neither in this study nor in previous ones was the effect of NT on modifications of body composition evaluated. Therefore, our aim in this study was to assess the impact of NT on body composition evaluated by CT scans in patients with gastric cancer prior to undergoing gastrectomy.

## 2. Materials and Methods

### 2.1. Patients and Study Design

A retrospective, observational, single-center study was performed. Inclusion criteria were a diagnosis of locally advanced GC undergoing gastrectomy (gastric cancer ≥ stage IB) who received NT before surgery, patients older than 18 years, available pre-surgical anthropometric data (weight and height), and two years of post-surgical follow-up. Patients with early tumor diagnosis, emergency surgery, resections with palliative intent and lack of subsequent follow-up were excluded.

Baseline patient characteristics and comorbidity were recorded. Pre-surgical comorbidity was classified according to the American Society of Anesthesiologists’ physical status classification. Tumor characteristics, pre-surgical supplementary nutrition and surgical parameters (surgical type resection, resection margins, surgery approach, type of lymphadenectomy, and positive nodes/total nodes ratio), and neoadjuvant dates were collected from the electronic medical records. Postoperative morbidity was recorded according to the Clavien–Dindo scale (grade of complications I–V).

### 2.2. Neoadjuvant Treatment

The inclusion criteria for NT in our unit were the following: resectable gastric cancer, stages II to III, patients under 85 years of age, an Eastern Cooperative Oncology Group (ECOG) test score from 0 to 1, and good renal, hepatic, and hematological function.

Until 2019, the MAGIC scheme (Cunningham et al. [2]) was used based on the combination of epirubicin, cisplatin, and 5-Fluorouracil. Subsequently, following the results of Al-Batran, the FLOT4 scheme (docetaxel, oxaliplatin, leucovorin, and fluorouracil for four preoperative cycles administered every two weeks, followed by radical surgery and four postoperative cycles with the same treatment schedule) was started [3]. Surgical intervention was scheduled at least four weeks after the last cycle of NT.

### 2.3. Study Composition by CT Scan

All images of each CT scan were extracted from the institutional PACS (Picture Archiving and Communication System) and transferred anonymously to the dedicated analysis tool (Synapse 3D). For the imaging study, the thinnest available slices (0.625–5 mm) of an axial section of the abdomen at the mid-level of the L3 vertebra were selected. Semi-automatic 2D segmentation was performed, and all measurements were taken from this radiological section.

Patients undergoing NT must have at least two radiological studies available and digitized. In patients with NT, the following two measurements were required: the CT scan before chemotherapy treatment was chosen for the first, and the re-evaluation CT scan at the end of chemotherapy (radiological study before surgery) was selected for the second. The radiological study closest to the surgical intervention was collected in patients without NT to compare these two groups.

The range of −29 to +150 Hounsfield Units (HU) was used to identify muscle tissue. The radiological diagnosis of sarcopenia was established based on the skeletal muscle mass index (SMMI), which was calculated as the sum of the cross-sectional areas at the L3 level of these muscles (cm^2^) and normalized to the square of the patient’s height (m^2^). The surface area of the psoas muscle index (PsoasI) was measured and normalized to the square of the patient’s height [(Right psoas + Left Psoas)/height^2^ (cm^2^/m^2^)].

Sarcopenia was defined based on the values published by Prado et al. [15] and later modified by Martin; for males with a BMI < 25 Kg/m^2^, a value of SMMI < 43 cm^2^/m^2^ was selected and for males with a BMI > 25 Kg/m^2^, an SMMI < 53 cm^2^/m^2^ was used; for females, an SMMI < 41 cm^2^/m^2^ was selected [16]. For PsoasI, there are no internationally recognized values.

Adipose tissue measurements were performed on the same images at the L3 level. The range −150 to −50 HU was used to identify visceral adipose tissue (VAT) and −190 to −30 HU for subcutaneous adipose tissue (SAT). Most Asian series choose for the diagnosis of obesity a VAT value > 100 cm^2^. There are few European series in this respect. However, given the significant anthropometric differences between Western countries and Eastern populations, we have selected the cut-off values published by the European series of Tegels VAT > 163.8 cm^2^ for males and VAT > 80.1 cm^2^ for females [9]. For SAT, there are no internationally recognized values.

### 2.4. Ethical Considerations

This study followed the Declaration of Helsinki on Ethical Principles of Medical Research and all subsequent amendments and modifications. In addition, approval was obtained from the Ethics Committee of our Centre (registration number 264/20). Due to the retrospective nature of this study, formal written consent was not required. Patients gave verbal informed consent to participate.

### 2.5. Sample Size Calculation

The sample size was calculated to detect a significant difference in the incidence of sarcopenia among patient groups, with a statistical power of 80% and a significance level of 0.05. Despite the lack of previous studies, it was determined that a sample size of *n* = 180–190 (in a 2:1 ratio between groups) would be necessary to detect a 15% difference in sarcopenia incidence between groups before surgery (after neoadjuvant).

### 2.6. Statistical Analysis

The Kolmogorov–Smirnov test was used to analyze the normality of quantitative variables. In the cluster analysis, continuous variables were compared using Student’s *t*-test, and categorical variables were analyzed, as appropriate, using the chi-square test or Fisher’s exact test (univariate analysis). Continuous variables were expressed as mean with standard deviation, and categorical variables were represented as proportions. Spearman correlation was used for the correlation of quantitative variable analysis.

Paired data statistical tests (the Wilcoxon signed-rank test and McNemar’s test) were used according to the nature of the variables. Body composition values were compared between pre-NT and pre-surgery measurements. A value of *p* < 0.05 was considered a statistically significant result. Data analysis was performed with the statistical software SPSS^®^ 25.0 for Windows (SPSS Inc., Chicago, IL, USA).

## 3. Results

### 3.1. Baseline Characteristics

In our electronic registry, 211 eligible patients underwent GC gastrectomy between January 2012 and December 2019. Finally, after selecting inclusion/exclusion criteria, of the 180 consecutive eligible patients, 61 received NT prior to gastrectomy (Figure 1).

The age of the global sample was 68.05 ± 9.1 years, with a slight predominance of males (55.7%). According to Lauren’s classification, the intestinal type was the predominant one (30 patients, 50%). The most frequent ASA category was ASA II (32 patients, 52.5%), followed by ASA III (27 patients, 44.3%). Patients without NT were older (*p* < 0.001), with similar sex distribution (*p* = 0.875) and similar preoperative anesthetic risk (*p* = 0.648). The most frequent tumor sites were distributed between the antrum (47.6%) and the body (32.8%). One in three patients underwent a minimally invasive approach (28 patients), and the most frequent surgery, by a slight advantage, was total gastrectomy with D2 lymphadenectomy. Analyzing the type of surgery, patients with NT had more total resections (*p* = 0.026) and laparoscopic procedures (*p* = 0.010), although with a higher incidence of surgical margin involvement (*p* = 0.018). Although patients without NT experienced more overall complications (49.6% vs. 27.9%), no statistically significant differences were observed when comparing them using the Clavien–Dindo scale (*p* = 0.871). Postoperative mortality and oncological stages were comparable between groups (*p* = 0.718 and 0.153, respectively) (Table 1).

### 3.2. Body Composition at Baseline

Before any interventions, the median body composition for patients undergoing NT was as follows: VAT 166.7 (164–244) cm^2^, SAT 131 (88–187.5) cm^2^, SMMI 46.69 (40.1–55.2) cm^2^/m^2^, and PsoasI 5.78 (4.5–7) cm^2^/m^2^. Before any treatment, body composition measurements, including the diagnosis of sarcopenia or obesity, were similar between groups, with no statistically significant differences identified (Table 2).

### 3.3. Body Composition Analysis after NT and before Surgery

Before undergoing gastrectomy, 61 patients received NT, and 119 underwent surgery directly.

Before starting any treatment, i.e., after the diagnosis of gastric cancer, the mediation of body composition and sarcopenia diagnosed 7.4% more patients with sarcopenia in the group that subsequently received neoadjuvant treatment. However, this difference between the two groups was not statistically significant. After receiving chemotherapy, patients in the neoadjuvant group worsened on the independent skeletal muscle measurement (SMMI) from a difference of −4.64% to −7.64% (Table 3). This loss in musculature translated into an 8.2% increase in sarcopenia diagnoses; that is, before surgery, patients with neoadjuvant treatment had 15.6% more sarcopenia than those without chemotherapy relapse (Figure 2).

A comparison was made on the variation in radiological body composition values before and after NT. An overall decrease in muscle mass assessment parameters was observed both in the quantification of total skeletal muscle mass (SMMI 46.69 (40.1–55.2) vs. 45.53 (39.5–51.7) cm^2^/m^2^) and for the PsoasI (5.78 (4.5–7) vs. 5.34 (4.5–6.1) cm^2^/m^2^). Likewise, according to the international definition of sarcopenia, after treatment with neoadjuvant chemotherapy, the diagnosis of sarcopenia increased overall by 8.20% (Table 4).

When analyzed by sex (Figure 2), there was a statistically significant worsening of muscle assessment parameters, mainly in men, who lost 5.55% of skeletal muscle mass and 10.5% when measured in the psoas muscle. Women showed a slight decrease in psoas muscle mass (1.31%) but not in the SMMI. In men, the incidence of sarcopenia increased from 29.4% pre-NT to 47.1% post-NT (*p* = 0.031). On the other hand, no significant changes were observed in visceral or subcutaneous adipose tissue either overall or by sex (Figure 3).

Furthermore, when analyzing not only the variations in numerical values of radiological parameters of body composition but also the percentage of patients who experienced a decline in muscle mass, it was observed that 82% of males lost some percentage of muscle mass during neoadjuvant therapy. In other words, only six patients increased their skeletal muscle mass during chemotherapy treatment (Figure 4).

## 4. Discussion

Our results show that patients who receive neoadjuvant therapy before gastrectomy for gastric cancer present deleterious changes in their body composition during treatment. Moreover, these changes mean that NT patients undergo surgery with a higher incidence of sarcopenia.

Currently, the benefit of NT on overall survival in GC has been demonstrated in several clinical trials [2,3]. For this reason, NT, followed by radical surgery, is the standard of care for GC [4,5]. On the other hand, sarcopenia and malnutrition are also very common in gastrointestinal malignancies, particularly those of the upper gastrointestinal tract [9]. Several studies have demonstrated the relationship between sarcopenia and poorer post-surgical and oncological outcomes in GC. The reported prevalence of sarcopenia in GC patients in the literature is variable and ranges from 12% to 69.8% [11,12]. In our sample, the global prevalence of sarcopenia was 35.6%. We observed that patients diagnosed with GC before treatment did not show significant differences in the analyzed characteristics. However, when both groups underwent surgery, NT patients presented significantly more sarcopenia (+15%) and less skeletal muscle mass.

Few studies have evaluated the specific setting of the impact of NT on body composition in patients undergoing gastrectomy for locally advanced GC, and all authors seem to agree that NT changes body composition [17,18,19,20]. After chemotherapy, our patients presented a significant global loss of muscle mass and a consequent increase in the prevalence of sarcopenia. Furthermore, we performed a subgroup analysis by sex and identified that muscle loss is unequal between men and women. Men had a muscle mass loss of 5%, whereas this did not occur in women, with the latter showing only a decrease in psoas muscle. Most authors cited above identified decreased muscle mass after NT [17,18,19,21,22]. However, only Horii et al. [18] and den Boer [20] suggest a possible sex difference. In fact, for Horii, the group they called the “remarkable psoas muscle loss group” included only men. Although, for this reason, a sex-specific analysis was not performed; the authors suggest that the poorer long-term prognosis of this group of patients is partly due to a sex difference [18]. On the other hand, Zhang et al. did not identify such muscle depletion with treatment [20]. The change in body fat with NT is a topic that not all authors explore in-depth, although, generally, there is a trend toward fat loss after chemotherapy.

The changes brought about by NT have been described in detail above. However, what is the translation of these bodily changes on overall survival? For example, a study in patients with GC and palliative chemotherapy has shown that marked loss of visceral and skeletal muscle adiposity significantly predicts worse overall and disease-free survival [23,24]. Along these lines, Zhang’s study concluded that low adipose tissue content is associated with poor long-term prognosis [20].

Our group recently published a study to analyze the oncological impact and postoperative complications of sarcopenia diagnosis in patients with gastric cancer prior to gastrectomy. In this study, it was identified that both sarcopenia and sarcopenic obesity were independent factors implicated in oncological prognosis as well as in postoperative morbidity and mortality. This group of patients had double the risk of experiencing severe postoperative complications (Clavien–Dindo > 3b) [14]. This same study also analyzed the contribution of neoadjuvant therapy as an independent factor for postoperative complications without demonstrating statistically significant differences. Moreover, the beneficial role of preoperative chemotherapy (as described in previous studies and cohorts, such as the one published by the Al-Batran group [3]) was not reflected in this cohort. One hypothesis in this regard, which ties the results of previous studies together, is that the high incidence of sarcopenia in the overall gastric cancer cohort makes the negative impact of sarcopenia so significant that the oncological benefit fails to reach significant differences.

In this study, like other studies [25], no significant differences were identified when specifically analyzing postoperative morbidity and mortality between groups. Furthermore, it was observed that perhaps the neoadjuvant group, although not statistically significant, exhibited less postoperative mortality. A relevant factor in this regard is the difference in age between the cohorts. Age is a risk factor associated with severe complications following gastrectomy for gastric cancer [26,27,28]. Additionally, aging is a factor associated with sarcopenia [29,30]. Perhaps for this reason, even though patients with neoadjuvant therapy (NT) faced surgery with a higher sarcopenic state (despite not presenting it before any treatment), the fact that they were statistically younger could have protected them from suffering morphological changes during treatment. These hypotheses need to be validated and further explored in future studies.

For these reasons, and despite the need for further studies to confirm these findings, gastric cancer patients (especially males) who require neoadjuvant chemotherapy prior to surgery modify their body composition, increasing the incidence of sarcopenia. Despite a significant increase in the diagnosis of sarcopenia in TN patients, this is not reflected in an increase in postoperative morbidity. Prospective studies are therefore needed to investigate the role of these changes and their impact on oncological prognosis.

Our study has some limitations, mainly that it is a retrospective study. For this reason, some aspects, such as nutritional supplementation, physical activity, or differences between races or socioeconomic effects, could not be evaluated. However, this is one of the first studies focusing on identifying, in a radiological and quantitative manner, the changes in muscle and fat mass experienced by patients undergoing NT for GC. Prospective studies are needed to identify the role of nutritional supplementation during NT and perioperatively for GC regarding its impact on body composition and sarcopenia.

## 5. Conclusions

Patients with locally advanced GC who receive NT have changes in body composition, especially in males, during chemotherapy. However, despite a higher incidence of preoperative sarcopenia, it was not associated with a significant increase in postoperative morbidity.

## Figures and Tables

**Figure 1 cancers-16-02408-f001:**
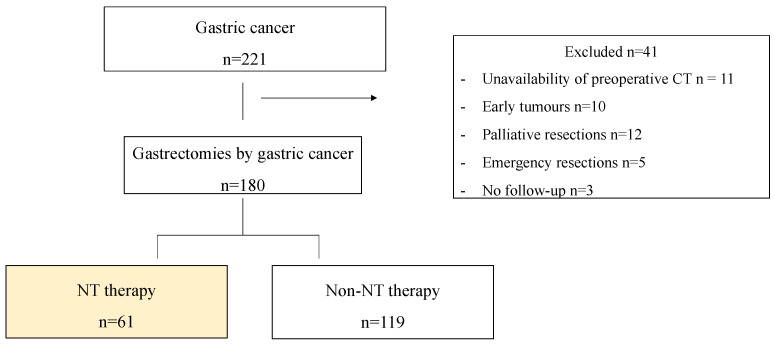
Flowchart of the study. NT: neoadjuvant chemotherapy; CT: computed tomography.

**Figure 2 cancers-16-02408-f002:**
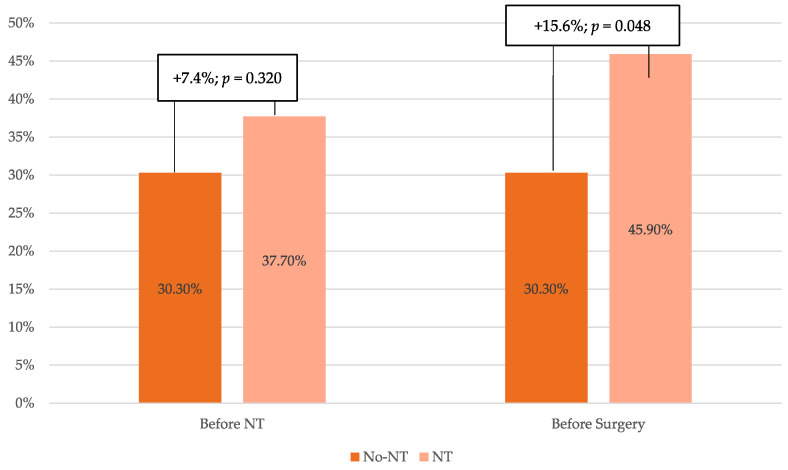
Variation in the diagnosis of sarcopenia before and after chemotherapy. NT: neoadjuvant chemotherapy.

**Figure 3 cancers-16-02408-f003:**
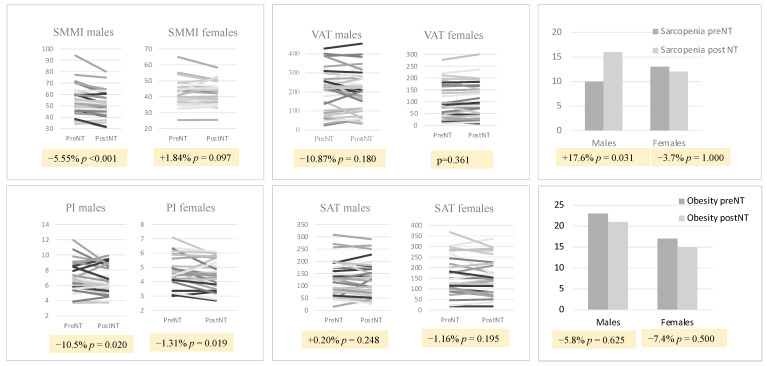
Variation in body composition values in patients with neoadjuvant therapy by sex. VAT: visceral adipose tissue; SAT: subcutaneous adipose tissue; SMMI: skeletal muscle mass index; PsoasI: Psoas Index; NT: neoadjuvant chemotherapy.

**Figure 4 cancers-16-02408-f004:**
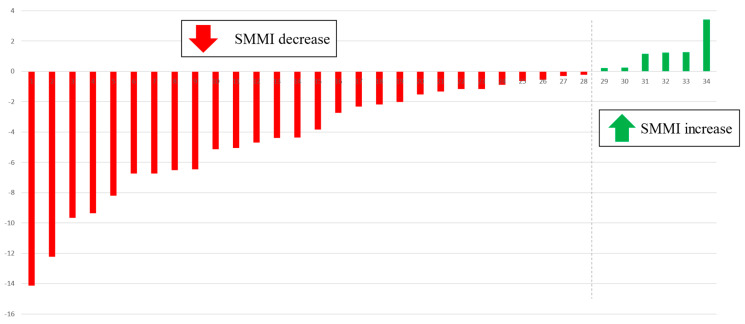
Waterfall plot chart of variation in SMMI in men after neoadjuvant treatment. Waterfall plot shows the difference between pre-neoadjuvant and post-neoadjuvant SMMI values in males (SMMI: SMMIneo cm^2^/m^2^). Each bar represents this difference in one patient. In 28 of the 34 males, this difference is negative with loss of muscle mass. SMMI: skeletal muscle mass index.

**Table 1 cancers-16-02408-t001:** Description and characteristics of patients by NT group.

Values	NT (*n* = 61)	Non-NT (*n* = 119)	*p* Value
Age (years)	68.05 ± 9.1	75.21 ± 11.2	*p* < 0.001
Sex (Male)	34 (55.7)	68 (57.1)	0.875
ASA classification			0.648
- I	2 (3.3)	7 (5.9)	
- II	32 (52.5)	60 (50.4)	
- III	27 (44.3)	50 (42)	
- IV	-	2 (1.7)	
Lauren’s type			0.080
- Intestinal	30 (50)	73 (63.5)	
- Signet ring cell	22 (36.7)	33 (28.7)	
- Other	8 (13.3)	9 (7.9)	
Tumor differentiation			0.105
- G1	10 (16.7)	24 (20.2)	
- G2	12 (20)	38 (31.9)	
- G3	38 (63.3)	55 (46.2)	
Tumor location			0.466
- Cardias	7 (11.5)	13 (10.9)	
- Fundus	3 (4.9)	4 (3.4)	
- Body	20 (32.8)	31 (26.1)	
- Antrum/pylorus	(47.6)	61 (51.3)	
- Whole stomach	2 (3.3)	2 (1.7)	
- Gastric stump	-	8 (6.7)	
Surgery (total gastrectomy)	43 (70.5)	63 (52.9)	0.026
Laparoscopic approach	28 (45.9)	32 (27)	0.010
Lymphadenectomy			0.456
- D1	2 (3.3)	15 (12.6)	
- D2	54 (88.5)	86 (72.2)	
- Other	5 (8.2)	18 (15.1)	
Positive surgical margins	9 (14.8)	5 (4.2)	0.018
Morbidity (overall)	17 (27.9)	59 (49.6)	0.007
Clavien–Dindo complications (90 days)			0.871
- I	1 (1.6)	7 (5.9)	
- II	8 (13.1)	24 (20.2)	
- III	4 (6.5)	18 (15.1)	
- IV	2 (3.2)	4 (3.4)	
Postoperative mortality (90 days)	2 (3.2)	6 (5)	0.718
Neoadjuvant scheme		-	-
- FLOT4	29 (47.6)		
- ECF	16 (26.2)		
- FOLFOX	6 (9.8)		
- EOX	5 (8.2)		
- CAPOX	5 (8.2)		
AJCC stage			0.153
- I	10 (16.4)	35 (29.4)	
- II	30 (49.2)	47 (39.5)	
-III	21 (34.4)	37 (31.1)	

ASA: American Society of Anesthesiologists; FLOT4: 5-fluorouracil, leucovorin, oxaliplatin and docetaxel-4; ECF: epirubicin, cisplatin and continuous 5-fluorouracil; FOLFOX 5: fluorouracil, leucovorin and oxaliplatin; EOX: epirubicin, oxaliplatin and capecitbine; CAPOX: oxaliplatin and capecitabine; AJCC: American Joint Committee on Cancer; IR: interquartile range; NT: neoadjuvant chemotherapy.

**Table 2 cancers-16-02408-t002:** Baseline body composition in NT and non-NT groups.

Values	Non-NT (*n* = 119)	NT (*n* = 61)	Difference	*p*
VAT (cm^2^)	167.8 (106–224)	166.7 (164–244)	−0.60%	0.383
SAT (cm^2^)	149.4 (108–202)	131 (88–187.5)	−1.83%	0.205
SMMI (cm^2^/m^2^)	49 (44–55)	46.69 (40.1–55.2)	−4.64%	0.139
PsoasI (cm^2^/m^2^)	5.54 (5–7)	5.78 (4.5–7)	+4.15%	0.975
Sarcopenia %	30.3 (36)	37.7 (23)	+7.4%	0.320
Obesity %	67.2 (80)	65.5 (40)	−1.7%	0.868

VAT: visceral adipose tissue; SAT: subcutaneous adipose tissue; SMMI: skeletal muscle mass index; PsoasI: Psoas Index; NT: neoadjuvant chemotherapy. Difference: Δ (NT − non-NT)/NT (%).

**Table 3 cancers-16-02408-t003:** Body composition before surgery.

Values	Non-NT (*n* = 119)	Post-NT (*n* = 61)	Difference	*p*
VAT (cm^2^)	167.8 (106–224)	153 (71–226)	–9.67%	0.753
SAT (cm^2^)	149.4 (108–202)	135.4 (76–177)	–10.33%	0.115
SMMI (cm^2^/m^2^)	49 (44–55)	45.52 (39–51)	–7.64%	0.027
PsoasI (cm^2^/m^2^)	5.54 (5–7)	5.34 (4–6)	–3.74%	0.261
Sarcopenia %	30.3 (36)	45.9 (28)	+15.6%	0.048
Obesity %	67.2 (80)	59 (36)	–8.2%	0.326

VAT: visceral adipose tissue; SAT: subcutaneous adipose tissue; SMMI: skeletal muscle mass index; PsoasI: Psoas Index; NT: neoadjuvant chemotherapy. Difference: Δ (Post-NT − non-NT)/Post-NT (%)

**Table 4 cancers-16-02408-t004:** Body composition changes in NT group.

Values	Pre-NT (*n* = 61)	Post-NT (*n* = 61)	Difference	*p*
VAT (cm^2^)	166.7 (164–244)	153 (71–226)	−8.95%	0.593
SAT (cm^2^)	131 (88–187.5)	135.4 (76–177)	+2.96%	0.119
SMMI (cm^2^/m^2^)	46.69 (40.1–55.2)	45.52 (39–51)	−2.57%	<0.001
PsoasI (cm^2^/m^2^)	5.78 (4.5–7)	5.34 (4.5–6.1)	−8.2%	0.002
Sarcopenia %	37.7 (23)	45.9 (28)	+8.2%	0.125
Obesity %	65.5 (40)	59 (36)	−6.5%	0.219

VAT: visceral adipose tissue; SAT: subcutaneous adipose tissue; SMMI: skeletal muscle mass index; PsoasI: Psoas Index; NT: neoadjuvant chemotherapy. Difference: Δ (Post-NT − Pre-NT)/Post-NT (%).

## Data Availability

Restrictions apply to the availability of the data generated or analyzed during this study to preserve patient confidentiality or because they were used under license. The corresponding author will, on request, detail the restrictions and any conditions under which access to some data may be provided.

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
