# Peer review of "Impact of Neoadjuvant Treatment on Body Composition in Patients with Locally Advanced Gastric Cancer"

_cancers, 2024, doi:10.3390/cancers16132408_

Round 1

Reviewer 1 Report

Comments and Suggestions for Authors

The findings presented in this study underscore the significant impact of neoadjuvant therapy (NT) on body composition, particularly in patients undergoing gastrectomy for gastric cancer. The observed increase in sarcopenia prevalence and muscle mass loss, especially among males, highlights a concerning trend that may affect surgical outcomes and overall prognosis.

The study effectively points out the potential implications of these changes on patient outcomes, suggesting that the nutritional status of patients, especially males, deteriorates during neoadjuvant chemotherapy, leading to surgery being conducted under suboptimal conditions. This underscores the importance of monitoring and addressing these body composition changes throughout the treatment process.

Additionally, the study acknowledges its limitations, particularly its retrospective nature, which may have implications for certain aspects such as assessing the impact of nutritional supplementation or exercise. However, it serves as a valuable contribution to the understanding of the effects of NT on body composition in gastric cancer patients and emphasizes the need for prospective studies to further elucidate the role of interventions such as nutritional supplementation in optimizing outcomes for these patients.

Overall this study is conducted well. The only minor comment is, this study is, it is conducted from retrospective data of single center. Did authors consider investigating racial aspect? I can understand this study was done in Spain and may be most of patients would be Spanish. Spanish life expectancy is generally higher 80’s range, which is way higher than other parts of world. Non-Hispanic blacks and women tend to experience more sarcopenia in general. It is very interesting that we do not see drastic body composition changes in female. Also, female life expectancy is higher than male in many parts of world.

Author Response

Thank you very much for your comments. 
The main part of the sample is Spanish and of Caucasian race. 
This possible bias has not been analysed but will be of great interest for future research. However, as it may be of interest, it has been added in the commentary on the limitations of the study. 

Reviewer 2 Report

Comments and Suggestions for Authors

A really interesting, well-organized, and clearly presented manuscript concerned a critical issue when patients with locally advanced gastric cancer are treated with neoadjuvant and its effects on body composition. I enjoyed reading the manuscript and found it interesting and helpful.

Comments on the Quality of English Language

A really interesting, well-organized, and clearly presented manuscript concerned a critical issue when patients with locally advanced gastric cancer are treated with neoadjuvant and its effects on body composition. I enjoyed reading the manuscript and found it interesting and helpful.

Author Response

Thank you very much for your kind words

Reviewer 3 Report

Comments and Suggestions for Authors

The authors present an interesting paper reviewing their incidence of sarcopenia and impact of NT on its development in gastric cancer patients. I have a few comments. 

1. The main value of the paper is to suggest that man undergoing NT are at serious risk of developing sarcopenia and recommendations to address and improve this body composition changes such as exercise therapy are necessary.

2. Regarding the analysis for the development of sarcopenia in Non-NT versus NT is very disingenuous; the threshold for significance was artificially set at 15% however there is not a 15 % development of sarcopenia during NT. There was a baseline difference in sarcopenia incidence 30% Non-NT versus 37.7% in the NT group,  and only a 7-8% development of sarcopenia after NT corresponding for the 15% difference before surgery, and one cannot attribute to NT the sarcopenia difference before initiation of NT. That has to be related with disease status, cancer stage, patient characteristics, etc.  Same exact analysis can be made to the SMMI, with a tru difference of 1.1.

3. The statistical power should have been to calculate the NAT group alone and the fact that difference can be seen with only 60 patients demonstrates the profound effect that treatment can have. More importantly it also speaks about gaps in treatment to have clinicians focus on exercise therapy during NT.

4. Lastly the authors clearly assert that sarcopenia is associated with higher post-operative complication. however, despite an almost 50% pre-surgical incidence of sarcopenia in the NT group  (15% more than in the non-NT group) the incidence of complications is about half of the incidence in the non-NT group, similar can be say about post surgical mortality. The authors have to explain this finding in some detail. 

Author Response

1. The main value of the paper is to suggest that man undergoing NT are at serious risk of developing sarcopenia and recommendations to address and improve this body composition changes such as exercise therapy are necessary.

Many thanks for the comments. The aim of the study was to look at variations in body composition in patients receiving NT. Certainly, although the study identifies differences, it does not actively propose strategies. However, identifying this risk group may be useful for future work and research. 

2. Regarding the analysis for the development of sarcopenia in Non-NT versus NT is very disingenuous; the threshold for significance was artificially set at 15% however there is not a 15 % development of sarcopenia during NT. There was a baseline difference in sarcopenia incidence 30% Non-NT versus 37.7% in the NT group,  and only a 7-8% development of sarcopenia after NT corresponding for the 15% difference before surgery, and one cannot attribute to NT the sarcopenia difference before initiation of NT. That has to be related with disease status, cancer stage, patient characteristics, etc.  Same exact analysis can be made to the SMMI, with a tru difference of 1.1.

Thank you very much for your comment. On reviewing the paragraph referred to we agree with the reviewer that it is not well expressed in the original text. What we wanted to refer to is that since there was no previous data on the subject, a 15% difference between groups (non-adjuvant vs. neoadjuvant) before surgery was considered significant. It is true that this is an artificial threshold but since there are no previous studies it was decided that a significant difference at this percentage would be sufficient to identify differences in radiological parameters. In fact, differences between groups were compared and statistically significant differences were found only with respect to age. We consider this to be logical given that in our centre one of the criteria for neoadjuvant treatment is age at diagnosis. We found no relative differences with respect to anatomopathological tumour stage.The reviewer's suggestion has been corrected in the text to improve comprehensibility. 

3. The statistical power should have been to calculate the NAT group alone and the fact that difference can be seen with only 60 patients demonstrates the profound effect that treatment can have. More importantly it also speaks about gaps in treatment to have clinicians focus on exercise therapy during NT.

Thank you again for your appreciation.  The main objective of the study was to determine how neoadjuvant patients face surgery with a higher incidence of sarcopenia and also to see if this difference occurred during treatment or was intrinsic to the oncological status of the disease. Therefore the sample size was calculated for the overall sample. It has been modified in the text to make it clearer. 

4. Lastly the authors clearly assert that sarcopenia is associated with higher post-operative complication. however, despite an almost 50% pre-surgical incidence of sarcopenia in the NT group  (15% more than in the non-NT group) the incidence of complications is about half of the incidence in the non-NT group, similar can be say about post surgical mortality. The authors have to explain this finding in some detail. 

I completely agree with the reviewer. It is true that no statistically significant differences in postoperative complications and/or mortality are identified in the work. That is not the objective of the study, so a regression analysis to identify factors associated with postoperative complications was not performed. In a previous study published by our group, it was identified that both sarcopenia and sarcopenic obesity were independent factors for experiencing severe complications (greater than Clavien-Dindo grade 3b postoperatively). However, having received neoadjuvant therapy was not an independent factor. This is similarly reflected in this work and is shown in Table 1.

Probably, the high incidence of sarcopenia in this group of patients makes the sarcopenic patients more susceptible to experiencing more complications, regardless of the preoperative chemotherapy received. To better delve into the topic, it has been further discussed in the discussion section.

Round 2

Reviewer 3 Report

Comments and Suggestions for Authors

The authors have addresses most of the comments in the manuscript, the main issue in my mind remains. The actual development of sarcopenia during NT is only about 8%, so the authors need to modify that table number 3. as it makes it like the groups were similar at baseline regarding sarcopenia and thew were not. The P value was not statistically significant but the 7.4% difference is not irrelevant. Moreover, one cannot say that the reason why this paper is important is because sarcopenia is "dangerous" for patients after surgery when your paper actually does not reflect that. A more reasonable conclusion is that the impact of NT in sarcopenia is modest and despite a possible increase in sarcopenia measurements surgical outcomes are not negatively impacted by NT

Author Response

The authors have addresses most of the comments in the manuscript, the main issue in my mind remains. The actual development of sarcopenia during NT is only about 8%, so the authors need to modify that table number 3. as it makes it like the groups were similar at baseline regarding sarcopenia and thew were not. The P value was not statistically significant but the 7.4% difference is not irrelevant. Moreover, one cannot say that the reason why this paper is important is because sarcopenia is "dangerous" for patients after surgery when your paper actually does not reflect that. A more reasonable conclusion is that the impact of NT in sarcopenia is modest and despite a possible increase in sarcopenia measurements surgical outcomes are not negatively impacted by NT

Many thanks to reviewer 3 for his comments. 

In accordance with his suggestions, a figure (Figure 3) has been added to explain more specifically the 8% increase in sarcopenia during neoadjuvant treatment. This figure explains that the difference already existed before any treatment was started and how the 8% increase makes the final difference 15%. In addition, an explanation has been added in the text. 

Similarly, and in accordance with your recollection, the discussion and conclusion of the study has been modified to emphasise that the effect was not reflected in postoperative morbidity.

We have also revised the English language and corrected different parts of the text. 

Thank you again for your feedback.

Round 3

Reviewer 3 Report

Comments and Suggestions for Authors

The authors addressed the concerns and the questions that arose during the review process